# Assessing the Sensitizing and Allergenic Potential of the Albumin and Globulin Fractions from Amaranth (*Amaranthus hypochondriacus*) Grains before and after an Extrusion Process

**DOI:** 10.3390/medicina55030072

**Published:** 2019-03-20

**Authors:** Feliznando Isidro Cárdenas-Torres, Cuauhtémoc Reyes-Moreno, Marcela de Jesús Vergara-Jiménez, Edith Oliva Cuevas-Rodríguez, Jorge Milán-Carrillo, Roberto Gutiérrez-Dorado, Jesús Gilberto Arámburo-Gálvez, Noé Ontiveros, Francisco Cabrera-Chávez

**Affiliations:** 1Food Science and Technology Post-Grade, Chemistry-Biological Sciences Faculty, University of Sinaloa, Culiacán 80019, Sinaloa, Mexico; feliznandoc@hotmail.com (F.I.C.-T.); edith.cuevas.r@uas.edu.mx (E.O.C.-R.); jmilanc@uas.edu.mx (J.M.-C.); robe399@hotmail.com (R.G.-D.); 2Nutrition Sciences Academic Unit, University of Sinaloa, Culiacán 80019, Sinaloa, Mexico; mjvergara@uas.edu.mx (M.d.J.V.-J.); fcabrera@uas.edu.mx (F.C.-C.); 3Department of Chemical and Biological Sciences, University of Sonora, Hermosillo 83000, Sonora, Mexico; gilberto.aramburo.g@gmail.com; 4Division of Sciences and Engineering, Department of Chemical, Biological, and Agricultural Sciences, University of Sonora, Navojoa 85880, Sonora, Mexico

**Keywords:** food allergy, amaranth, extrusion, murine model, sensitization

## Abstract

*Background:* The first cases of food allergy to amaranth grain have recently been published. This pseudocereal is considered hypoallergenic, and there is scarce information about the allergenic potential of amaranth proteins, either before or after food processing. *Objective:* To evaluate, in a mouse model of food allergy, the sensitizing and allergenic potential of extruded and non-extruded albumin and globulin fractions from amaranth grains. *Materials and Methods:* Amaranth (*Amaranthus hypochondriacus*) flour was obtained and the albumin and globulin fractions isolated. These protein fractions were also obtained after flour extrusion. An intraperitoneal 28-day protocol was carried out to evaluate the sensitizing and allergenic potential of the proteins. The common and rarely allergenic proteins ovalbumin and potato acidic phosphatase were utilized as reference. Specific IgE and IgG antibodies were evaluated for all the proteins tested. Mast cell protease-1 (mMCP-1) responses were evaluated in serum samples collected after intragastric challenges with the proteins of interest. All serological evaluations were carried out using ELISA. *Results:* Mice were sensitized to the non-extruded albumin fraction from amaranth grains and to ovalbumin (*p* = 0.0045). The extrusion process of amaranth proteins abrogated the IgE responses triggered under non-extruded conditions (*p* = 0.0147). mMCP-1 responses were significantly detected in the group of mice sensitized to ovalbumin (*p* = 0.0138), but not in others. *Conclusions*: The non-extruded albumin fraction from amaranth has the potential to sensitize BALB/c mice, but this sensitizing potential fails to induce detectable serum levels of the mast cell degranulation marker mMCP-1 after intragastric challenges. Furthermore, the extrusion process abolished the sensitization potential of the amaranth albumins.

## 1. Introduction

Food allergy is defined as a reproducible adverse reaction to food which is mediated by immunological mechanisms. This disorder is the most common cause of allergic reactions, either in children or adults [1]. Individuals are sensitized once B cells produce allergen-specific IgE antibodies and these IgEs are bound by their Fc region to FcεRI receptors present on the cell surface of mast cells [2,3]. Upon re-exposure to the relevant allergen, two adjacent allergen-specific IgEs on the surface of mast cells could crosslink with the allergenic protein, and this triggers the degranulation of the cells, which releases mediators of the allergic response, such as histamine and tryptase, and mast cell protease-1 (MCP-1) [4]. Certainly, sensitization per se is not enough to trigger the symptoms associated with allergic reactions, but is essential for potential IgE-mediated allergic disease.

Food allergy affects 1%–2% of the adult population [5,6] with peanuts, tree nuts, egg, soybeans, fish, crustacea, milk, and wheat being the main allergens [7]. On the contrary, amaranth is considered hypoallergenic, and it is used to replace other commonly allergenic grains in foodstuff formulations. Particularly, amaranth has been used in some Latin American countries as an ingredient to produce muesli bars (traditionally called “alegrías”), but in other regions, it has become popular as an ingredient for industrialized goods (e.g., breakfast cereals, bread, pasta, milk surrogates, and protein isolates, among others) [8]. However, a few cases of amaranth allergy have been reported [9,10,11]. Although profilin from pollen of *Amaranthus* plants can trigger episodes of respiratory allergies [12,13], this is not the relevant allergen in allergy cases triggered by amaranth ingestion. Evidence suggests that other unidentified proteins with molecular mass greater than profilin are the triggers of allergy reactions in such cases [9]. Notably, recent studies have shown the ability of a murine model to discriminate between common and rarely allergenic proteins [14], and this allows the evaluation of the sensitizing and allergenic potential of proteins before and after food processing.

Certainly, the molecular conformation of proteins could change after food processing, modifying their allergenic characteristics [15]. For instance, some studies have shown that the extrusion process could reduce the number of conformational epitopes in flours containing allergens such as peanuts and hazelnuts [16,17], but other studies have reported no effect of extrusion on the allergenicity of proteins [18]. Amaranth proteins could be used as ingredients to give texture and improve the nutritional quality of extruded food products. However, the impact of food processing, such as extrusion, on the sensitizing and allergenic potential of amaranth proteins remains uncertain. Thus, our aim was to evaluate the effect of the extrusion process on the sensitizing and allergenic potential of the albumin and globulin fractions from amaranth in a BALB/c mouse model of food allergy.

## 2. Materials and Methods

### 2.1. Amaranth Flour and Extrusion Process

Amaranth grains (*Amaranthus hypochondriacus*) were purchased at a local market in Temoac, Morelos, Mexico. The grains were cleaned, and amaranth flour was obtained. For the extrusion process, batches of the flour (400 g) were nixtamalized with calcium hydroxide (0.21% (w/w)) and distilled water (28% (w/w)) [19]. The extrusion was carried out in a single-screw extruder model 20DN (CW Brabender Instruments, Inc., New York, NY, USA). The extrusion conditions were screw speed of 124 rpm; moisture = 28%; constant temperature = 130 °C [19]. The extruded product was collected, tempered, dried, and milled (UD Cyclone Sample Mill, UD Corp., Boulder, CO, USA) until a particle size of 0.25 mm was obtained.

### 2.2. Protein Extraction

Proteins were obtained using an extraction kit for total vegetable proteins, following the manufacturer’s instructions (Sigma-Aldrich, St. Louis, MO, USA, Cat. PE0230). A sequential extraction of albumins and globulins was carried out according to the Osborne classification [20]. Briefly, the flour was resuspended in acetone in a 1:10 (w/v) ratio and incubated at 4 °C for 16 h with continuous stirring. Subsequently, the samples were centrifuged (16,000× *g*, 20 min, 4 °C) and the pellets collected and dried with air at room temperature (12 h). Afterward, the samples were resuspended (100 mg/mL) in distilled water (1 h, continuous stirring) and centrifuged (16,000× *g*, 20 min, 4 °C). The supernatants containing the albumin fraction were collected and the pellets resuspended (100 mg/mL) in 0.05 mol/L Tris-HCl, (pH = 8, 1 h, continuous stirring). After centrifuging as previously described, the supernatants containing the globulin fraction were collected. Both albumin and globulin fractions were dialyzed against distilled water in 1 kDa membranes (Spectrum Laboratories, Rancho Dominguez, CA, USA, Cat. 132105) and lyophilized.

### 2.3. Ovalbumin and Potato Acid Phosphatase Preparations

Ovalbumin (OVA, grade V ≥98% pure, Sigma-Aldrich, Cat. A5503) and potato acid phosphatase (PAP, 0.5–3.0 unit/mg solid, Sigma-Aldrich, P3752) were used as allergenic and hypoallergenic proteins, respectively [21,22]. The proteins were dissolved in PBS (pH 7.4) in order to obtain a 0.2 mg/mL OVA solution and 2.0 mg/mL PAP solution, as previously described [14]. All protein concentrations were measured using the bicinchoninic acid method following the manufacturer’s instructions (BCA assay, Thermo Scientific Pierce^TM^, Rockford, IL, USA).

### 2.4. SDS-PAGE Analysis

Total protein and the albumin and globulin fractions were electrophoretically separated in 15% polyacrylamide gels under denaturing and reducing conditions (SDS-PAGE). Wide Range Molecular Weight Markers (Sigma-Aldrich, Cat. S8445, 6500−200,000 Da) were used, and 20 µg of test protein were loaded per lane. The electrophoresis was run in a Mini-PROTEAN^®^TGX system (Bio-Rad, Hercules, CA, USA) at an initial voltage of 70 V during the first 10 min and at 200 V for additional 2 h. A commercial running buffer was used (Bio-Rad, Cat. 1610732). The gels were stained using Bio-Safe Coomassie stain (Bio-Rad, Cat. 161-0786), and the resulting gel image was digitalized.

### 2.5. Sensitization Protocol and Ethical Aspects

Four- to five-week-old female BALB/c mice (*n* = 6 per group) were used. All animals (a total of 72) were free of pathogens and were fed a diet free of proteins from amaranth, egg, or potato for at least two generations. Water and food were available ad libitum. The room temperature was maintained at 25 °C, 50% ± 10% relative humidity, and 12 h light/dark cycle. The ethics review board of the Nutrition Sciences Academic Unit of the Autonomous University of Sinaloa approved the study protocol (Ethical approval number: CE-UACNyG-2014-JUL-001).

The mice were sensitized as previously described [23]. Briefly, different concentrations of amaranth proteins (0.25, 0.05, or 0.025 mg) were administered intraperitoneally in a final volume of 0.25 mL of PBS, pH 7.4. The treatments were repeated on days 3, 6, 9, and 12 after the first immunization (day 0) (Figure 1). The control groups received PBS only. The mice were bled on days 0, 28, and 35, and serum samples obtained and stored at −80 °C until their use. Taking into account the doses that triggered the highest anti-non-extruded albumin and globulin IgE response (0.05 and 0.25 mg per mouse, respectively), the albumin and globulin fractions from extruded amaranth were administered. OVA and PAP were administered as previously described (0.05 and 0.5 mg per mouse, respectively) [14].

Following sensitization, an intragastric challenge was carried out on day 35. For this purpose, a solution of the relevant protein (0.25 mL at 10 mg/mL) was administered using plastic feeding tubes (Instech Laboratories, Inc., Cat. 20 GA × 38 mm). After 30 min of challenge, the mice were bled from the tail vein, and serum samples were obtained and stored at −80 °C until their use (Figure 1).

### 2.6. Enzyme-Linked Immunosorbent Assay

Specific IgE and IgG antibodies were evaluated using ELISA. Briefly, 96-well microplates (NUNC Maxisorb., Waltham, MA, USA, Cat. 442404) were coated with 100 μL of antigen (200 μg/mL) in coating buffer (BioLegend, Cat. 421701) and incubated overnight at 4 °C. The plates were washed (three times, 200 μL per well) with PBS/Tween-20 0.05% pH 7.4 and incubated for 2 h at room temperature with 10% fetal calf serum in PBS pH 7.4. After three washes, 100 μL of serum samples diluted 1:10 (IgE) and 1:1000 (IgG) in 10% fetal calf serum were added into the wells and the plates incubated overnight at 4 °C. The plates were washed, as previously described, and 100 μL of biotinylated rat anti-mouse IgEa antibody (BioLegend, Cat. 408804) diluted 1:250 in 10% fetal calf serum (final concentration of 2 μg/mL) was added to each well and incubated for 1 h at room temperature. For the detection of IgG antibodies, a rat anti-mouse IgG antibody was used (BioLegend, Cat. 405303). After washing, 100 µL of streptavidin-horseradish peroxidase diluted 1:1000 in ELISA diluent (BioLegend, Cat. 405210) was added to each well, and the plates were incubated at room temperature for 30 min. Finally, after six washes, 100 μL of tetramethylbenzidine (TMB, Thermo Scientific, Cat. 34028, Rockford, IL, USA) was added and the plates incubated in the dark to allow color development. The reaction was stopped with 50 μL of 2 N H_2_SO_4_. The time for color development was 3 min for IgG and 30 min for IgE. The absorbance was measured at 450 nm in a microplate reader (Thermo Scientific, Cat. 51119000). All sera were analyzed in duplicate.

### 2.7. mMCP-1 Serum Evaluation

Mouse mastocyte protease 1 (mMCP-1) was evaluated with a commercially available ELISA kit (BioLegend, Cat. 432702) following the manufacturer’s instructions. The reaction was stopped with 50 μL of 2 N H_2_SO_4_. The time for color development was 15 min and the absorbance was read at 450 nm. The concentrations of mMCP-1 in the serum samples were calculated using standard curves.

### 2.8. Statistical Analysis

Data were analyzed using GraphPad Prism version 6.0 (GraphPad Software, San Diego, CA, USA). In order to test the normality of the data, the Kolmogorov–Smirnov test was used. For multiple comparisons, ANOVA and Tukey tests were used. Kruskal–Wallis test and Dunn’s test were used to compare the data that failed in normality. Normally distributed data were expressed as mean and standard deviation. Non-normally distributed data were expressed as median and interquartile range. A *p*-value < 0.05 was considered significant.

## 3. Results

### 3.1. SDS-PAGE

Figure 2 shows the electrophoretic pattern of the total protein and protein fractions from non-extruded and extruded amaranth. The electrophoretic pattern of total proteins remained consistent before and after extrusion (Figure 2, lanes 1 and 2). However, the pattern of the albumin and globulin fractions changed after the extrusion process (Figure 2, lanes 3–6). Overall, albumin and globulin fractions from non-extruded amaranth showed more protein bands than their extruded counterparts. The protein bands between 24 and 29 kDa, which are visible in the albumins from non-extruded amaranth, were not detected after the extrusion process. In the globulin fraction, most protein bands between 14.2 and 45 kDa were not detected in the globulin fraction from extruded amaranth.

### 3.2. Albumins, But Not Globulins, Trigger Consistent IgE Responses in BALB/c Mice

IgE responses were detected in all mice that were administered the non-extruded albumin fraction from amaranth, and this was independent of the protein dose administered (Figure 3a). Compared to the control group, the responses were statistically significant for all the protein concentrations evaluated (*p* < 0.05). On the contrary, no IgE responses were detected in the groups of mice that were administered the globulin fraction from amaranth (*p* > 0.05) (Figure 3b). Based on these results, and considering the available proteins, further studies were carried out using the concentrations of 0.05 and 0.25 mg of amaranth proteins.

### 3.3. Amaranth Protein Fractions, Ovalbumin, and Potato Acid Phosphatase Trigger IgG Responses

In order to verify that all the proteins tested were immunologically relevant, the IgG responses were evaluated. As expected, no IgG responses were detected in pre-immune sera (Figure 4; day 0). Contrary and compared to the control group, significant IgG responses were detected in all the groups of mice immunized with the different proteins tested (*p* < 0.05) (Figure 4; day 28).

### 3.4. The Extrusion Process Abolishes the IgE Immune Response Triggered by Amaranth Albumins

Consistent with previous experiments carried out in this study, the mice that underwent the 28-day protocol of sensitization, and were administered the non-extruded albumin fraction from amaranth, triggered an IgE immune response (*p* < 0.05, compared to the other proteins tested, except OVA) (Figure 5). This response was not detectable in a comparable group of mice that was administered the extruded albumin fraction from amaranth (Figure 5). As expected, OVA triggered the most robust IgE immune responses (*p* < 0.05, compared to the other proteins tested) (Figure 5). No IgE immune responses were detected after the administration of the remaining proteins tested (Figure 5).

### 3.5. Mice Sensitized to Amaranth Albumins Fail to Trigger mMCP-1 Responses

In order to evaluate the effector phase of food allergy, the mastocyte degranulation marker mMCP-1 was assessed. After an intragastric challenge, only the group of mice sensitized to OVA showed detectable levels of mMCP-1 (*p* < 0.05, compared to the other groups) (Figure 6). All the other groups failed to trigger mMCP-1 responses, even the group that was administered the non-extruded albumin fraction from amaranth (Figure 6), a fraction that showed the potential to trigger a weak and consistent IgE immune response readily detectable using ELISA (Figure 3a and Figure 5).

## 4. Discussion

In this study, the sensitizing and allergenic potential of the albumin and globulin fractions from non-extruded and extruded amaranth were evaluated. The SDS-PAGE analysis of proteins showed that there were no changes in the electrophoretic pattern of the total amaranth proteins extracted either before or after extrusion, but the pattern of the albumin and globulin fractions changed after the extrusion process. These discrepancies in the electrophoretic patterns can be attributed to the methods of protein extraction employed. On the one hand, total amaranth proteins were extracted using a method that involves chaotropic and reducing compounds, which ensure increased solubility of proteins. On the other hand, the albumin and globulin fractions were extracted using either water for albumins, or Tris-HCl (pH = 8) for globulins, without the use of chaotropic and reducing compounds [20]. Overall, the results support the notion that some amaranth proteins could lose their solubility in aqueous solutions after extrusion processing [24].

We and others have shown that the dose of allergen, as well as the number and frequency of immunizations, are key factors to sensitize BALB/c mice to food proteins [14,25,26]. The adjuvant-free 28-day protocol with five immunizations used in the present study was shown to be effective to detect even allergens that hardly trigger IgE immune responses after intraperitoneal administration [14]. The globulin fraction from amaranth was tested at three doses, but no anti-globulin IgE responses were detected either under extruded or non-extruded conditions. By contrast, robust anti-globulin IgG immune responses were triggered, highlighting that the doses of protein tested were immunologically relevant. These results could be limited by the fact that, in ELISA evaluations, low titers of allergen-specific IgE antibodies could be masked by the presence of allergen-specific IgGs [14]. Although the serum samples were not tested after IgG depletion, there was no increase of mMCP-1 in the sera of mice collected after intragastric challenge with the globulins of interest. Thus, it can be hypothesized that amaranth globulins are unlikely to trigger allergic disease after oral exposure due to a lack of, or quite low, sensitizing potential.

The non-extruded globulin fraction from amaranth triggered an IgE immune response. This response was weak compared to the anti-OVA IgE response, but consistent among all the independent experiments carried out. Based on our results, the characteristics of the albumin allergenic epitopes that give rise to sensitization in BALB/c mice remain uncertain. Conformational epitopes are unlikely to trigger allergic disease in food allergy cases since these epitopes are degraded after ingestion by gastric or intestinal enzymes [27]. Linear epitopes are thought to be the main epitopes implicated in food allergy. However, intragastric challenge of mice sensitized to non-extruded albumin failed to increase the serum levels of the mastocyte degranulation marker mMCP-1. Our results show that amaranth albumins have the potential to sensitize and are more likely than globulins to trigger allergic reactions after amaranth ingestion.

Contrary to the non-extruded albumin fraction from amaranth, the extruded fraction failed to trigger an IgE immune response. Furthermore, it failed to increase the serum levels of mMCP-1 after intragastric challenge with the protein of interest. The lack of anti-albumin IgE responses could be attributed to the loss of allergenic epitopes as a consequence of the extrusion process, since some protein bands were missed after extrusion. Previous studies have shown that the extrusion process negatively impacts the sensitizing and allergenic potential of food allergens, either reducing the IgE immune responses triggered or lowering the serum levels of mMCP-1 [16,17]. Taken together, under the basis of the sensitizing and allergenic potential evaluations in a murine model, the results show that amaranth albumins and globulins can be considered as hypoallergenic ingredients in foodstuff formulations, especially when the food processing involves an extrusion process.

Finally, we should acknowledge that our study has some limitations. Firstly, the mouse model used has to be better validated although it was shown to be effective in discriminating between common and rarely allergenic proteins, such as OVA and PAP. Secondly, due to evidence suggesting that cases of food allergy to amaranth exist [9,10,11], studies on amaranth allergy patients including skin-prick tests and oral challenges are required to make definitive conclusions about the allergenic potential of the albumin and globulin fractions from amaranth. Besides these limitations, our study contributes to understanding the impact of food processing on the allergenicity of proteins and serves as evidence in the search of a validated animal model to evaluate the inherent sensitizing and allergenic potential of proteins.

## 5. Conclusions

The non-extruded albumin fraction from amaranth has the potential to trigger weak anti-albumin IgE responses. However, this sensitizing potential is not enough to detect, in serum samples, the mastocyte degranulation marker mMCP-1 after intragastric challenges with amaranth albumins. Notably, the extrusion process abolished the sensitizing potential of amaranth albumins. This study suggests that amaranth could be used as a surrogate for other recognized allergenic grains, such as wheat and soy.

## Figures and Tables

**Figure 1 medicina-55-00072-f001:**
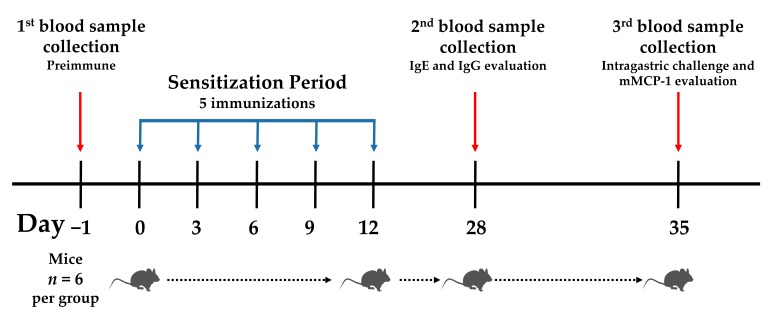
Sensitization protocol and intragastric challenge.

**Figure 2 medicina-55-00072-f002:**
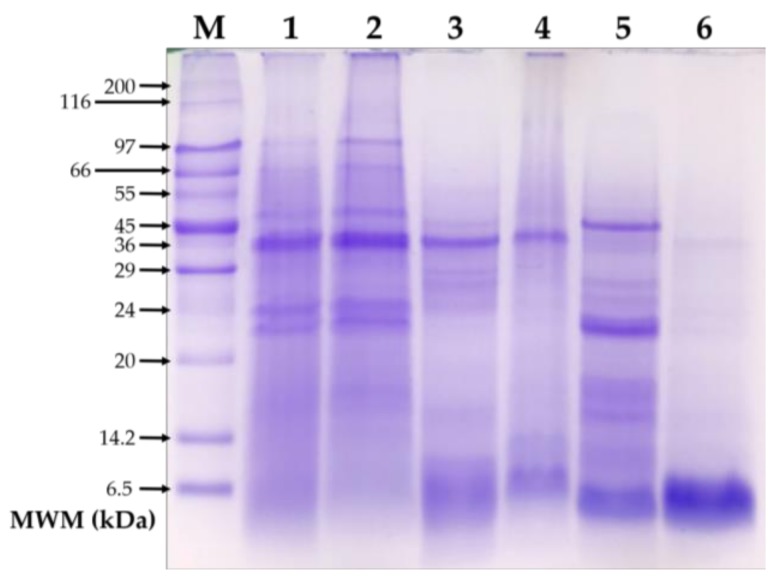
Electrophoretic pattern of proteins. **M**: Wide Range Molecular Weight Marker; **1**: Total protein of non-extruded amaranth; **2**: Total protein of extruded amaranth; **3**: Albumin fraction of non-extruded amaranth; **4**: Albumin fraction of extruded amaranth; **5**: Globulin fraction of non-extruded amaranth; **6**: Globulin fraction of extruded amaranth.

**Figure 3 medicina-55-00072-f003:**
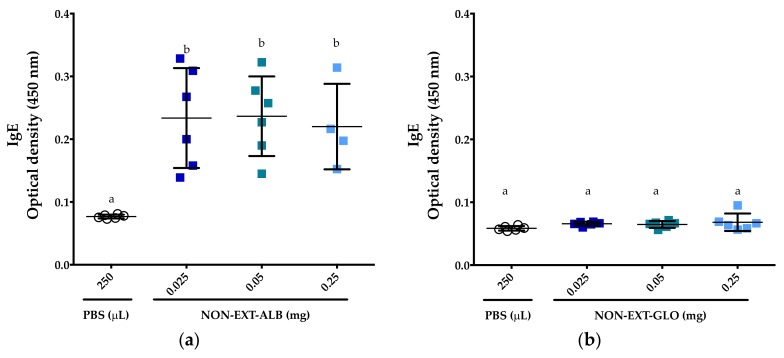
IgE responses to albumins and globulins. (**a**) Responses to the albumin fraction from non-extruded amaranth (NON-EXT-ALB); (**b**) Responses to the globulin fraction from non-extruded amaranth (NON-EXT-GLO). All serum samples tested were collected after the mice underwent a 28-day protocol of sensitization. PBS: Phosphate-buffered saline. Comparisons in each figure were carried out using ANOVA/Tukey tests. Vertical bars indicate standard deviations. Different letters indicate significant differences (*p* < 0.05).

**Figure 4 medicina-55-00072-f004:**
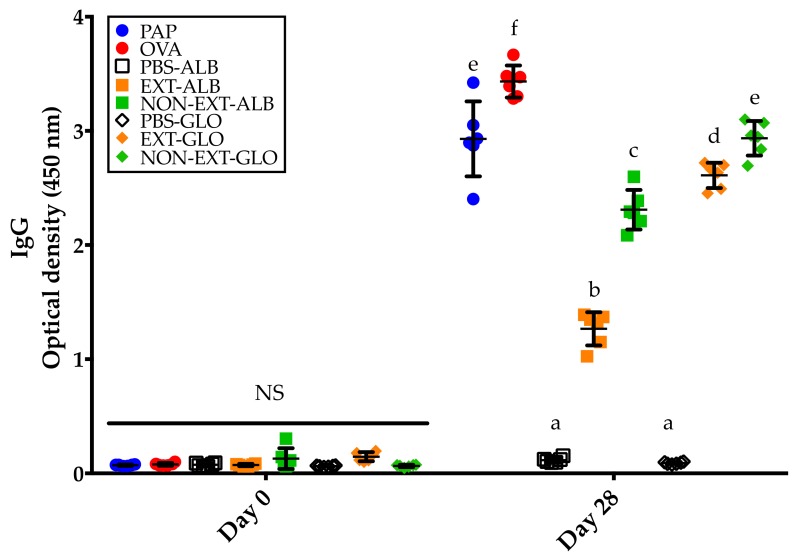
IgG responses to the different proteins tested. The evaluations were carried out in serum samples collected before and after the mice underwent the 28-day sensitization protocol. PAP: Potato acidic phosphatase (0.5 mg); OVA: Ovalbumin (0.05 mg); PBS-ALB: Phosphate-buffered saline as control for albumins; EXT-ALB: Extruded albumins (0.05 mg); NON-EXT-ALB: Non-extruded albumins (0.05 mg); PBS-GLO: Phosphate-buffered saline as control for globulins; EXT-GLO (0.25 mg): Extruded globulins; NON-EXT-GLO: Non-extruded globulins (0.25 mg). Comparisons at day 0 and 28 were carried out separately using ANOVA/Tukey tests. Vertical bars indicate standard deviations. Different letters indicate significant differences (*p* < 0.05).

**Figure 5 medicina-55-00072-f005:**
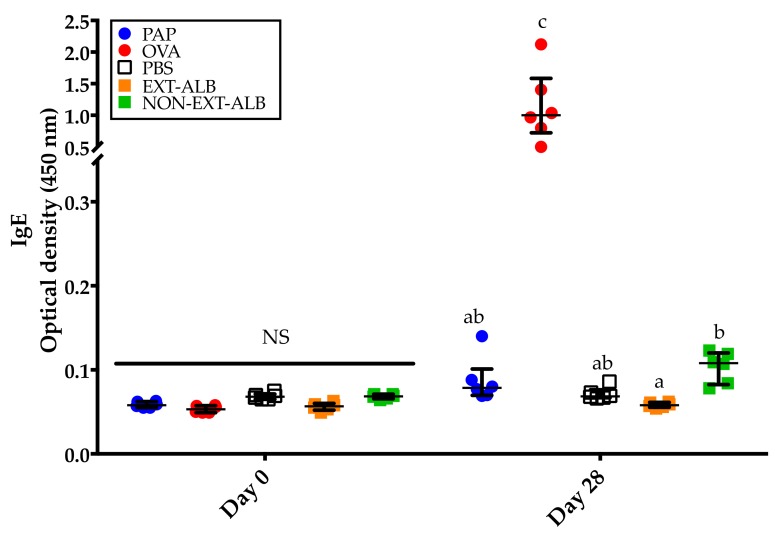
IgE responses to different proteins tested. The evaluations were carried out in serum samples collected before and after the mice underwent the 28-day sensitization protocol. PAP: Potato acidic phosphatase (0.5 mg); OVA: Ovalbumin (0.05 mg); PBS: Phosphate-buffered saline; EXT-ALB: Extruded albumins (0.05 mg); NON-EXT-ALB: Non-extruded albumins (0.05 mg). Comparisons at day 0 and 28 were carried out separately using Kruskal–Wallis/Dunn tests. Vertical bars indicate interquartile ranges. Different letters indicate significant differences (*p* < 0.05).

**Figure 6 medicina-55-00072-f006:**
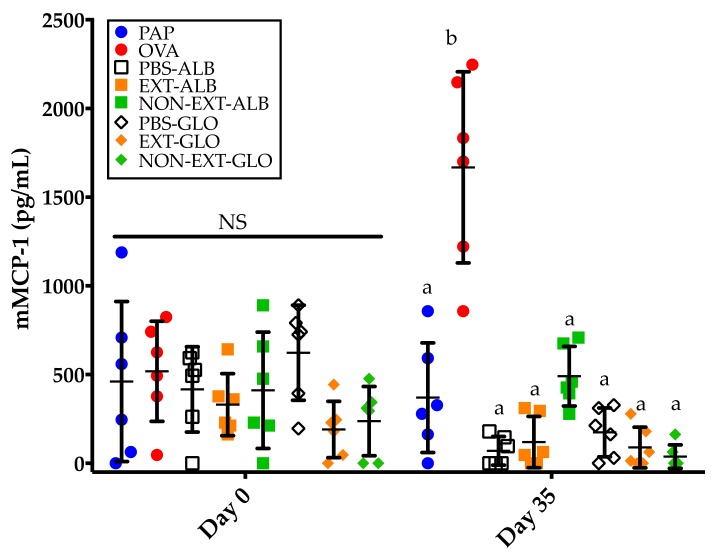
Serum mMCP-1 concentrations before (day 0) and after (day 35) intragastric challenges (2.5 mg). PAP: Potato acidic phosphatase (0.5 mg); OVA: Ovalbumin (0.05 mg); PBS-ALB: Phosphate-buffered saline as control for albumins; EXT-ALB: Extruded albumins (0.05 mg); NON-EXT-ALB: Non-extruded albumins (0.05 mg); PBS-GLO: Phosphate-buffered saline as control for globulins EXT-GLO: Extruded globulins (0.25 mg); NON-EXT-GLO: Non-extruded globulins (0.25 mg). Comparisons at day 0 and 35 were carried out separately using ANOVA/Tukey tests. Vertical bars indicate standard deviations. Different letters indicate significant differences (*p* < 0.05).

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
