# Peer review of "Assessing the Sensitizing and Allergenic Potential of the Albumin and Globulin Fractions from Amaranth (Amaranthus hypochondriacus) Grains before and after an Extrusion Process"

_medicina, 2019, doi:10.3390/medicina55030072_

Round 1

Reviewer 1 Report

1. Abstract: Conclusions:"The unprocessed albumin fraction from amaranth grains has the potential to sensitize, but degranulation of mast cells could not occur upon oral reexposition to the proteins." The authors need to rewrite to clarify the conclusions in the abstract. The conclusions section of the paper is well written but the conclusions section of the abstract is poorly written and not clear.The authors should replace this sentence in the abstract with modified statements from the conclusions section of the paper.

2. Please replace re-exposition by "re-exposure" throughout the manuscript.

3. Introduction- line 53 and 54- "Food allergy affects 1-2% of the adult population [5,6] being peanuts, tree nuts, egg, soybeans, 53 fish, crustacea, milk, and wheat the main allergens...

Grammatically incorect- please correct to- "Food allergy affects 1-2% of the adult population [5,6] with peanuts, tree nuts, egg, soybeans, 53 fish, crustacea, milk, and wheat being the main allergens...."

4. line 54- "Contrary, amaranth is considered..."Please add "On the" before contrary

5. lines 68 and 69-"However, it remains uncertain the impact of food processing such as extrusion on the sensitizing and allergenic potential of amaranth proteins". Grammatically incorrect. Replace by- "However, the impact of food processing such as extrusion on the sensitizing and allergenic potential of amaranth proteins remains uncertain."

6. line 142: "added to each well and incubated 1 h at room temperature".. Please add "for" after incubated. 

7. line 166- The electrophoretic pattern of total proteins remains consistent. Replace remains by "remained"

8. line 182-....Contrary, no IgE responses....Add "On the" before contrary

9. line 183-...Based on these results and considering the available of proteins, further studies were carried out using the concentrations of 0.05 mg and 0.25 mg of amaranth proteins. Please delete "of" after available

Author Response

Thank you very much for your review. Please find attached (PDF) the responses to your comments and suggestions. 

Reviewer 2 Report

Review Report

The paper focuses on characterization of allergens that can potentially elicit IgE-mediated food allergic reactions. Authors provide a nice summary of IgE mediated food allergic reactions and common allergenic foods in introduction and further describe Amaranthus as a food having low allergenic potential.  Despite considered as a hypoallergenic food, a case of food allergy to amaranthus has been reported. The objective of study is to characterize albumin and globulin allergenic proteins in pre and post extrusion and further use a mouse model to study the allergenic potential of these proteins.

Strengths

1.    The objective of study i.e. to characterize the allergenic protein in amaranthus, which is considered a hypoallergenic food is very relevant, and adds new information on this topic.

2.    The authors have provided a detailed summary of their study design in methods, statistical methods used.

3.    The figures are very descriptive and summarizes the study design and results appropriately

4.    The final conclusions is consistent with results drawn from data analysis that unprocessed allergens from amaranthus have potential to generate IgE sensitivity, but not to cause significant mast cell mediated allergic reactions.

Limitations

1.    Can consider adding a line about use of amaranthus as a cereal/food in introduction, as it is not a commonly used food in most parts of world. This will generate more interest for readers.

2.    No other major limitations or weaknesses

Specific Comments-

#53-#54- Food allergy affects 1-2% of the adult population [5,6] being peanuts, tree nuts, egg, soybeans, 53 fish, crustacea, milk, and wheat being the main allergens. – remove being from line 53 and add it after wheat.

#56 - However, the first cases of amaranth allergy have been reported. Can consider changing it to either to the first case or “a few cases”.

Author Response

(The authors gave the same response as above.)
